# Catalytic Reaction Mechanism of NO–CO on the ZrO_2_ (110) and (111) Surfaces

**DOI:** 10.3390/ijms20246129

**Published:** 2019-12-05

**Authors:** Xuesong Cao, Chenxi Zhang, Zehua Wang, Xiaomin Sun

**Affiliations:** 1Environment Research Institute, Shandong University, Qingdao 266200, China; xuesongcao@mail.sdu.edu.cn (X.C.); sdzhw@mail.sdu.edu.cn (Z.W.); 2College of Biological and Environmental Engineering, Binzhou University, Binzhou 256600, China; sdzhangcx@gmail.com

**Keywords:** Nitric oxide, Carbon monoxide, Zirconium dioxide, Density functional theory, Reaction mechanism

## Abstract

Due to the large population of vehicles, significant amounts of carbon monoxide (CO), nitrogen oxides (NO_x_), and unburned hydrocarbons (HC) are emitted into the atmosphere, causing serious pollution to the environment. The use of catalysis prevents the exhaust from entering the atmosphere. To better understand the catalytic mechanism, it is necessary to establish a detailed chemical reaction mechanism. In this study, the adsorption behaviors of CO and NO, the reaction of NO reduction with CO on the ZrO_2_ (110) and (111) surfaces was performed through periodic density functional theory (DFT) calculations. The detailed mechanism for CO_2_ and N_2_ formation mainly involved two intermediates N_2_O complexes and NCO species. Moreover, the existence of oxygen vacancies was crucial for NO reduction reactions. From the calculated energy, it was found that the pathway involving NCO intermediate interaction occurring on the ZrO_2_ (110) surface was most favorable. Gas phase N_2_O formation and dissociation were also considered in this study. The results indicated the role of reaction intermediates NCO and N_2_O in catalytic reactions, which could solve the key scientific problems and disputes existing in the current experiments.

## 1. Introduction

Heterogeneous catalysis always be used to decrease the emission of automobile exhausts (CO, NO_x_ and HC), such as the Rhodium catalytic system to convert toxic gas NO to inactive product N_2_ [1,2]. However, the rare and expensive characteristics of noble metal Rh limit its widespread use. Thus, quantities of investigations have been conducted to develop effective and economical catalyst systems.

As zirconium dioxide (ZrO_2_) has high thermal stability, excellent redox properties and an acid–basic site on its surface, it is a good catalyst and support material for various reactions, such as CO_2_ methanation [3,4], water–gas shift [5,6], NH_3_ selective catalytic reduction [7,8], and hydrodeoxygenation [9,10]. Especially, the transition metal oxide dispersed onto the surface of ZrO_2_ exhibits powerful activity for NO reduction. Okamoto et al. [11] found that Cu/ZrO_2_ catalysts showed high NO conversion to N_2_ at low temperature through a nitrous oxide (N_2_O) intermediate for a NO–CO reaction. As the supporter, ZrO_2_ has better performance than CeO_2_ in reducing the energy barrier of NO dissociation [12]. Besides, ZrO_2_ also provides abundant NO_x_ adsorption sites [13,14,15], which is beneficial for NO reduction. Koga et al. discovered that c-ZrO_2_ (110) ultrathin film covering a Cu surface exhibited high NO_x_ reduction activity [16].

NO reduction by CO may simultaneously eliminate two kinds of pollutants and convert them into harmless CO_2_ and inactive N_2_, via the stoichiometric reaction 2CO + 2NO → 2CO_2_ + N_2_. Dramatically, isocyanate (NCO) species and gas phase N_2_O were detected via experimental equipment during the reaction of NO with CO [17,18]. NO–CO reaction mechanisms involving NCO and N_2_O as intermediate complex have been considered on Co_3_O_4_ (110)-B [19], Pd/γ-Al_2_O_3_ (110) [20], and Cu-doped SrTiO_3_ (100) [21] catalyst surfaces. It is widely believed that CO abstracts lattice oxygen of catalyst to produce CO_2_ and oxygen vacancy through the Mars–van Krevelen mechanism [22], and then NO reduction occur on the defective catalyst surface via NCO or N_2_O intermediate species to form N_2_. Although previous studies have provided significant insight on NO–CO catalytic reaction mechanism, the catalytic cycle paths of NO reduction by CO on ZrO_2_ surfaces remain elusive.

Surface formation energies of three low-index c-ZrO_2_ (100), (110) and (111) were computed, as shown in Appendix A. In this study, we adopted the most stable surface (111) and a relatively reactive surface (110) as the exposed surfaces. The adsorption behaviors of CO and NO on ZrO_2_ (110) and (111) surfaces and possible reaction pathways for CO_2_ and N_2_ formation with corresponding energy evolution were carefully discussed based on DFT calculations. Through in-depth analysis and research, the role of reaction intermediates NCO and N_2_O in catalytic reactions will be clarified, which can solve the key scientific problems and disputes existing in the current experiments.

## 2. Results and Discussion

### 2.1. CO and NO Adsorption on ZrO_2_ (110) Surface

The structure models of ZrO_2_ (110) surface, as shown in Figure 1a, exposes threefold coordinated O (O_3c_) and sixfold coordinated Zr (Zr_6c_) atoms. In the surface, the Zr–Zr bond distance is 3.710 Å and the Zr–O bond distance is 2.247 Å. The lattice constant of ZrO_2_ bulk is 5.115 Å, which is in good agreement with experimental result of 5.090 Å [23].

Three adsorption sites were considered (Figure 1b): (1) the top site of Zr (Zr_T_), (2) the top site of O (O_T_), and (3) the bridge site between two O atoms (O_b_). We have systematically calculated the adsorption energies of CO (C-end and O-end) and NO (N-end and O-end) gas molecules on ZrO_2_ (110) surface, and the Table 1 lists all the value of *E*_ads_.

As shown in Figure 1, after fully structural optimization, CO (C-end and O-end) and NO (N-end and O-end) gas molecules preferred to adsorb on Zr_T_ site of ZrO_2_ (110) surface. As for CO molecule adsorption, *E*_ads_ of CO (C-end and O-end) adsorbed on Zr_T_ site was −0.676 eV and −0.360 eV, respectively, indicating the C-end adsorption was more energetically favorable than O-end. In the same way NO N-end adsorption was more stable than NO O-end adsorption on ZrO_2_ (110) surfaces. Coordinates for all the optimized structures were presented in Appendix A.

### 2.2. Reaction Mechanism of NO Reduction with CO on ZrO_2_ (110) Surface

#### 2.2.1. Path 1

The Mars−van Krevelen (MvK) mechanism is a universal reaction step for CO oxidation [24] and NO–CO reaction [19] on metal oxide surfaces. As shown in Figure 2, the NO–CO catalytic cycle reaction started when the first CO gas molecule adsorbed on Zr_T_ site (state *ii*) with an adsorption energy of −0.676 eV, and with the change of lattice O–C distance to 3.098 Å. After adsorption, CO extracted surface oxygen atom to form CO_2_ and a surface oxygen vacancy (state *iii*) through the MvK mechanism. In the corresponding transition state (TS1), the lattice O-CO bond decreased to 1.568 Å and the lattice O–C–O angle changed to 116.105°. Moreover, the energy barrier for this process was 1.948 eV, the result clearly shows that CO oxidation reaction can happen smoothly on ZrO_2_ (110) surface by using exhaust temperature. We calculated two cycle reaction mechanisms (path 1 and path 2) and N_2_O formation mechanism all share common steps to form oxygen vacancies on ZrO_2_ (110) surface.

In the next step, CO_2_ desorption into the gas phase (state *iv*) costs energy of 0.283 eV, then O_V_ was occupied by the first NO gas molecule forming a O_V_-NO complex structure (state *v*), and the energy of 2.310 eV is released. Subsequently, the next CO weakly adsorbed to adjacent O atom of the NO (state *vi*) with an adsorption energy of −0.510 eV. In TS2, the N-O bond broke and O atom moved toward CO to form OC-O bond by overcoming an energy barrier of 2.263 eV, and the OC-O bond length was changed from 2.683 Å at state *vi* to 1.637 Å. After TS2, the second CO_2_ molecule formation occurs (state *vii*). The reaction was exothermic by 1.246 eV. The similar reaction steps have been explored for Pd/γ-Al_2_O_3_ (110), the second CO abstracted lattice O atom from N-loaded Pd/γ-Al_2_O_3_ (110) surface with an energy barrier of 1.88 eV [20].

Then, the second CO_2_ desorbed into the gas phase leaving a N-doped surface (state *vii*), in which N atom embedded at surface oxygen vacancy site. The desorption energy of CO_2_ is approximately 0.3 eV, the interaction of ZrO_2_ surface with CO_2_ was much weaker than that with NO (*E*_ads_ ≈ 0.7 eV), which have been reported by Luo et al. [13]. After the second CO_2_ desorption, the second NO gas molecule located at N-doped ZrO_2_ (110) surface (state *ix*) with the binding energy of −0.068 eV. Once NO interacted with the embedded N atom, formation of a relatively stable intermediate complex bent N_2_O (state *x*) was very easy. As we can see from the energy profile (Figure 2), the process could facilely occur without energy barrier and release 3.286 eV energy. The energy barrier for N_2_O dissociation was extremely low only 0.002 eV. After TS3, N-O bond and N-Zr bond dissociated to produce N_2_ (state *xi*), then N_2_ desorption into gas phase cost 0.458 eV, O atom successively diffused to oxygen vacancy site via an obvious barrierless process with releasing energy of 0.510 eV, the integrated ZrO_2_ (110) surface was recovered eventually.

Accordingly, from calculated energy profile (Figure 2), the overall catalytic cycle is strongly exothermic. However, the second CO oxidation has greater activation energy of 2.263 eV, determining the rate of catalytic cycle. We proposed the other catalytic cycle path to produce CO_2_ and N_2_ via NCO species.

#### 2.2.2. Path 2

As presented in Figure 3, the reaction mechanism of O_V_ surface formation is shared with path 1. Subsequently, NO adsorbed on O_V_ site, the value of NO O-end adsorption was −1.605 eV, which bound weaker than that of NO N-end. Then, the second CO directly combined with N atom to generate the NCO intermediate (−1.614 eV, state *vi*). The second NO adsorbed on Zr_T_ site adjacent to NCO complex (−0.208 eV, state *vii*). In the co-adsorption configuration, NCO and NO moved toward each other via TS2 to form a NNCO_2_ four-membered ring intermediate (state *viii*) with an energy barrier of 0.355 eV. The subsequent step was NNCO_2_ intermediate dissociation to produce N_2_ and CO_2_ (state *ix*) with a barrier of only 0.141 eV. With N_2_ and CO_2_ desorption by costing 0.563 eV energy, the catalytic surface was recovered.

As we can see from the energy profile (Figure 3), the rate-determining step is the CO oxidation by the lattice oxygen, indicting ZrO_2_ (110) surface exhibits weakly catalytic activity for low-temperature oxidation of CO. Liang et al. observed that c-ZrO_2_ ultrafine powder showed relatively high activity for CO oxidation with light-off temperature at ~280 °C (50% conversion) and ~550 °C (100% conversion) [25,26]. As reported in the literature, metal doping was particularly effective for the formation of oxygen vacancy [21,27]. As for NO reduction reaction, the formation of oxygen vacancy is crucial for NO adsorption and reduction reaction [28,29,30]. On defective ZrO_2_ (110) surface, NO decomposition and N_2_ formation are occur easily as a result of a very small barrier of 0.355 eV via path 2. Although for the defective Co_3_O_4_ (110)-B surface [19], the most favorable NO reduction processes are energetically less competitive with a higher barrier of 1.48 eV, i.e., the ZrO_2_ (110) surface is remarkable for NO reduction.

#### 2.2.3. N_2_O formation.

Beginning with the oxygen vacancy surface (state *iv*, Figure 4) the first gas NO molecule located at O_V_ site (state *v*, Figure 4) by releasing 1.605 eV energy, the second incoming NO molecule N-end bound with surface N atom (state *vi*, Figure 4) with strong exothermicity of 2.872 eV. Subsequently, the ONN-O bond broke into N_2_O via TS2 (Figure 4). After TS2, the gas-phase N_2_O (state *vii*, Figure 4) formation, the barrier for the process was 0.987 eV, the reaction was exothermic by 1.345 eV. Eventually, N_2_O desorption into the gas phase (state *viii*, Figure 4) cost 0.456 eV and ZrO_2_ (110) surface recovered.

In cold start engines or lean-burn conditions N_2_O is the main by-product of NO reduction [31], and the formation of N_2_O on ZrO_2_ surface has been observed through NO temperature-programmed desorption experiment [13]. N_2_O is one of the six greenhouse gases specified in the Kyoto Protocol, the greenhouse activity is 310 times of CO_2_ [32]. Catalytic systems for N_2_O decomposition should be employed in catalytic converters.

Three reaction pathways have been calculated on the ZrO_2_ (110) surface: the gas phase N_2_O was observed via experimental equipment and in [18] it was also formed by theoretical calculation; our work provides the mechanism of N_2_O formation. Path 1 involving bent N_2_O intermediate was proposed to produce harmless CO_2_ and inactive N_2_, this cycle had a relatively high barrier than path 2 involving surface NCO intermediate, indicting CO_2_ and N_2_ formation mainly through path 2. Besides, the activation barriers for N_2_O and N_2_ formation on O_V_-ZrO_2_ (110) surface were 0.987 eV and 0.355 eV, respectively. The results imply that NO is selectively converted to N_2_ versus N_2_O under mild conditions.

### 2.3. CO and NO Adsorption on ZrO_2_ (111) Surface

ZrO_2_ (111) surface consists of 3-fold-coordinated oxygen atoms (O_3c_) and 7-fold-coordinated zirconium atoms (Zr_7c_), as shown in Figure 5a.

In Figure 5b three possible adsorption sites are labeled: (1) the top site of Zr (Zr_T_); (2) the top site of O (O_T_); (3) the 3-fold O-hollow site (O_H_). The value of adsorption energies is listed in Table 2.

For the adsorption behavior of CO (C-end and O-end) and NO (N-end and O-end) on ZrO_2_ (111) surface, we again researched that the Zr_T_ site was the most favorable adsorption site; the CO C-end adsorbed on Zr_T_ site had maximum adsorption energy −0.788 eV, which suggested that CO C-end adsorption was slightly preferred on ZrO_2_ (111) surface. Moreover, NO O-end adsorption was thermodynamically impossible, because of the positive *E*_ads_.

### 2.4. Reaction Mechanism of NO Reduction with CO on ZrO_2_ (111) Surface

#### 2.4.1. Path 1’

Based on the lowest energy structures of CO and NO on ZrO_2_ (111) surface, the first step was gas-phase CO interaction with surface Zr atom (−0.410 eV, state *ii*), as shown in Figure 6, CO may incorporate lattice O via TS1 where the distance between CO molecule and O atoms was decreased from 2.787 Å to1.246 Å. After TS1, a gas phase CO_2_ molecule (state *iii*) formed. The energy barrier was equivalent to 2.949 eV. Next, CO_2_ overcame 0.130 eV binding energy desorption into gas-phase. Specifically, the following path 1’ and N_2_O formation mechanisms share as common steps as discussed above.

Note that the energy barrier corresponding to TS1 (2.949 eV, Figure 6) is evidently high and oxygen vacancy formation on ZrO_2_ (111) surface is difficult at low-temperature. Once oxygen vacancy formation on ZrO_2_ (111) surface, NO adsorption and reduction are facile.

In state *v*, the NO molecule N-end was located at O_V_ site with a strong exothermicity of 2.923 eV. Subsequently, the second CO bonded with O atom of the NO to form a NOCO complex (−0.107 eV, state *vi*). The second CO_2_ formation (state *vii*) arose from NOCO complex with an energy barrier of 0.405 eV (TS2). Desorption of CO_2_ cost 0.134 eV. Then, the second NO adsorption on the N-doped surface (−0.629 eV, state *ix*) via TS3 forms the intermediate N_2_O (state *x*) with a small barrier of 0.350 eV. Followed by a barrierless process of N_2_O decomposition to generate N_2_ (state *xi*). With the N_2_ desorption, the surface was recovered.

#### 2.4.2. Path 2′

As shown in Figure 7, after the formation of oxygen vacancy, the O_V_ site was naturally replenished by NO molecule O-end (−3.056 eV, state *v*). The second CO approached toward N atom of adsorbed NO (−0.382 eV, state *vi*) to generate the NCO intermediate (state *vii*) through TS2) and an energy barrier of 0.385 eV was needed. The next NO could adsorb close to NCO complex (−1.582 eV, state *viii*) to realize the formation of four-membered ring intermediate NNCO_2_ (state *ix*). The activation barrier for this process was 1.194 eV. Subsequently, the N–C bond and N–O bond cleavage of four-membered ring NNCO_2_ led to CO_2_ and N_2_ formation (state *x*), with an almost barrierless processes (0.006 eV). Desorption of CO_2_ and N_2_ cost 0.255 eV.

#### 2.4.3. Mechanism involving N_2_O

The formation process of N_2_O was same as the case found on ZrO_2_ (110) surface. The second NO bonded with N atom of first adsorbed NO. With the N-O and N-Zr bond dissociation, N_2_O formation. The energy profile and corresponding structure models of reactant, transition state and product were illustrated in Figure 8.

As N_2_O is a hazardous by-product, its decomposition processes have been conducted on transition metal surfaces [33,34]. Following with the decomposition mechanism of N_2_O on Pd-O_V_/γ-Al_2_O_3_ (110) surface proposed by Gao [20]. We calculated N_2_O decomposition process on O_V_-ZrO_2_ (111) surface, as shown in Figure 9. The binding energy of N_2_O adsorption was equivalent to −0.166 eV, in the corresponding transition state the N–O bond length increased from 1.195 Å to 1.288 Å. By breaking the already activated N–O bond, the desired product N_2_ was formed and the dissociated O atom filled the O_V_. The reaction barrier for N_2_O decomposition is 0.288 eV, and the N_2_ desorption from the ZrO_2_ (111) surface requires 0.128 eV.

From calculated energy barriers, we found that CO oxidation by surface lattice oxygen was the rate-determining step during the process of NO reduction with CO, the reaction barriers were 1.948 eV and 2.949 eV on ZrO_2_ (110) and (111) surfaces, respectively. The results indicated that compared with ZrO_2_ (111) surface, ZrO_2_ (110) surface had more remarkable ability to catalyze NO reduction with CO because of the lower activation barrier. From the calculated adsorption energies, CO and NO all preferred to adsorb on ZrO_2_ (110) surface, which is consistent with the results of surface formation energies for (110) surface was relatively reactive than (111) surface. For the reaction mechanism of NO reduction by CO, similar reaction processes were found on ZrO_2_ (110) and (111) surfaces. Intermediate complex bent N_2_O was produced during path 1, path 1’ and → NCO intermediate was produced during path 2, path 2’, hazardous N_2_O gas molecule was formed both on ZrO_2_ (110) and (111) surfaces.

## 3. Materials and Methods

Periodic DFT calculations were executed using DMol^3^ module of Material Studio software package [35,36]. The exchange-correlation functional was treated by generalized gradient approximation with the Perdew-Wang 91 (GGA-PW91) [37]. The double numerical plus d-functions (DND) basis set was used to optimize all spin unrestricted structures. SCF tolerance was employed to 1.0 × 10^−5^, the convergence tolerance of maximum energy change, maximum force, and maximum displacement were set as 2.0 × 10^−5^ Ha, 0.004 Ha/Å, and 0.005 Å, respectively. The transition states (TS) were calculated with linear synchronous transit (LST) and quadratic synchronous transit (QST) method [38,39] and vibrational analysis was performed to identify TS have only one imaginary frequency.

The slab models of ZrO_2_ (110) and (111) surfaces were built with a 15–20 Å vacuum thickness to avoid interaction between neighboring slabs. The ZrO_2_ (110) surface consisting of four atomic layers with the lowest two layers being kept fixed in their bulk positions, whereas the others were allowed to relax with adsorbed molecules. The Brillouin zone was sampled with a (1 × 2 × 1) Monkhorst–Pack k-point grid [40]. The ZrO_2_ (111) surface consisting of six atomic layers with the bottom four layers were kept fixed and the top two layers were relaxed. A (2 × 2 × 1) Monkost–Pack grid [40] for the ZrO_2_ (111) surface geometry optimization. In all cases, an orbital cutoff of 5.0 Å was used to improve the computational performance and a 2 × 2 supercell was applied.

The adsorption energies (*E*_ads_) of CO and NO gas molecules on ZrO_2_ (110) and (111) surfaces were calculated by the following formula,
E_ads_ = E _(surface + gas molecules)_ − E _(surface)_ − E _(gas molecules)_(1)
where *E*
_(surface + gas molecules)_ is the total energy of the system involving the ZrO_2_ (110) or (111) catalyst with the adsorbed CO or NO gas molecule, *E*
_(surface)_ is the total energy of isolated ZrO_2_ (110) or (111) surface, and *E*
_(gas molecules)_ is the total energy of a single CO or NO gas molecule.

To validate the methods, we calculated the energy of NO, CO, N_2_ and CO_2_ molecules in a large unit cell of 10 Å × 10 Å × 10 Å, and compared the bond distances with the values calculated at the B3LYP/6-31G(d) level [41]. We listed all the values in Appendix A, the calculated bond distances were within 1% of ones.

## 4. Conclusions

In summary, DFT calculations were performed to investigate NO reduction with CO on ZrO_2_ (110) and (111) surfaces. For the ZrO_2_ (110) surface, two cycle reaction pathways (paths 1 and 2) for CO_2_ and N_2_ formation were presented, and the path 2 involving the process of the NCO intermediate interaction with NO was energetically favorable. The similar cycle reaction pathways (paths 1’ and 2’) were found on ZrO_2_ (111) surface, but CO oxidation by lattice O had a significantly higher energy barrier of 2.949 eV. The results indicated that compared with ZrO_2_ (111) surface, ZrO_2_ (110) surface had more remarkable ability to catalyze NO reduction with CO. Our calculations also clearly showed that the existence of oxygen vacancies on ZrO_2_ (110) and (111) surfaces were crucial for NO adsorption and reduction reaction. NO healing O_V_ was a strongly exothermic process (2.0–3.0 eV) and successive reactions all exhibited lower energy barriers, especially path 2 (0.355 eV) and path 1’ (0.405eV). The mechanism involving N_2_O also was explored, and the activation barriers for N_2_O formation on O_V_-ZrO_2_ (110) and (111) were 0.987 eV and 0.515 eV, respectively. The relatively higher barriers imply that NO is selectively converted to N_2_ versus N_2_O under mild conditions.

## Figures and Tables

**Figure 1 ijms-20-06129-f001:**
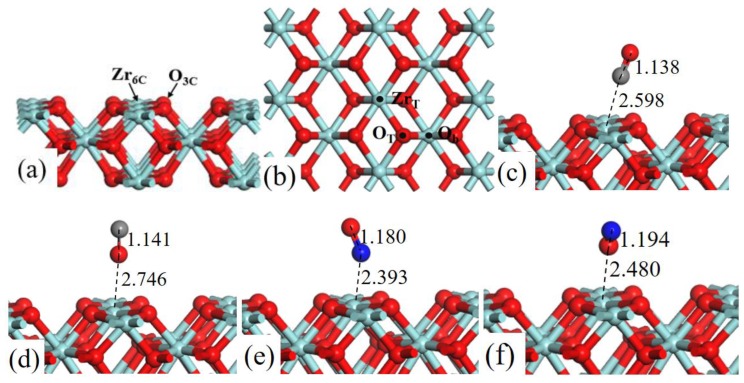
Structure models of (**a**) the side view and (**b**) corresponding top view of ZrO_2_ (110)-2 × 2 surface. (b) Three possible adsorption sites for CO and NO gas molecules are labeled. Optimized adsorption structures of (**c**) CO C-end, (**d**) CO O-end, (**e**) NO N-end, and (**f**) NO O-end on Zr_T_ site. Red, cyan, gray, and blue spheres represent the O, Zr, C, and N atoms, respectively.

**Figure 2 ijms-20-06129-f002:**
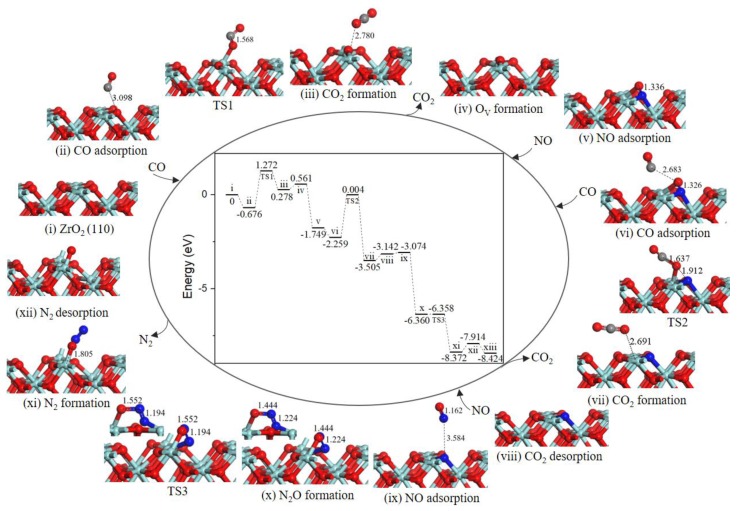
Energy profile and structure models of reactants, transition states and products for path 1 (NO N-end embed in oxygen vacancy site on ZrO_2_ (110) surface). The energy profile is placed in the center, and structure models are placed around.

**Figure 3 ijms-20-06129-f003:**
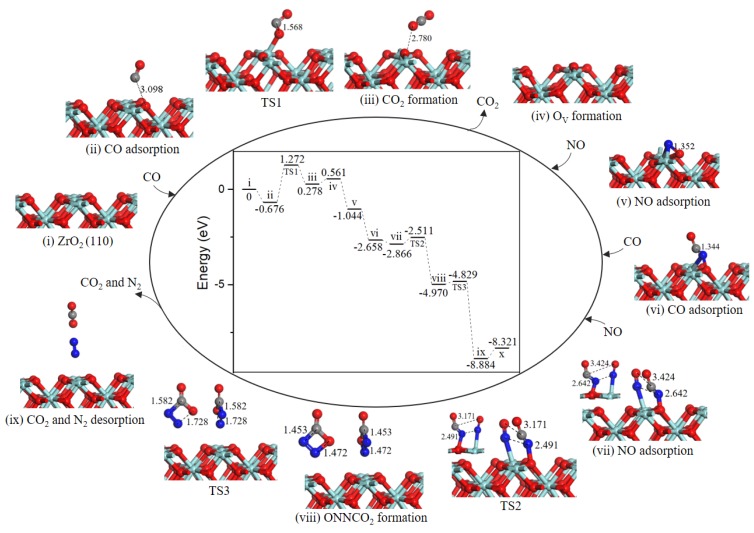
Energy profile and structure models of reactants, transition states and products for path 2 (NO O-end embed in oxygen vacancy site on ZrO_2_ (110) surface). The energy profile is placed in the center, and structure models are placed around.

**Figure 4 ijms-20-06129-f004:**
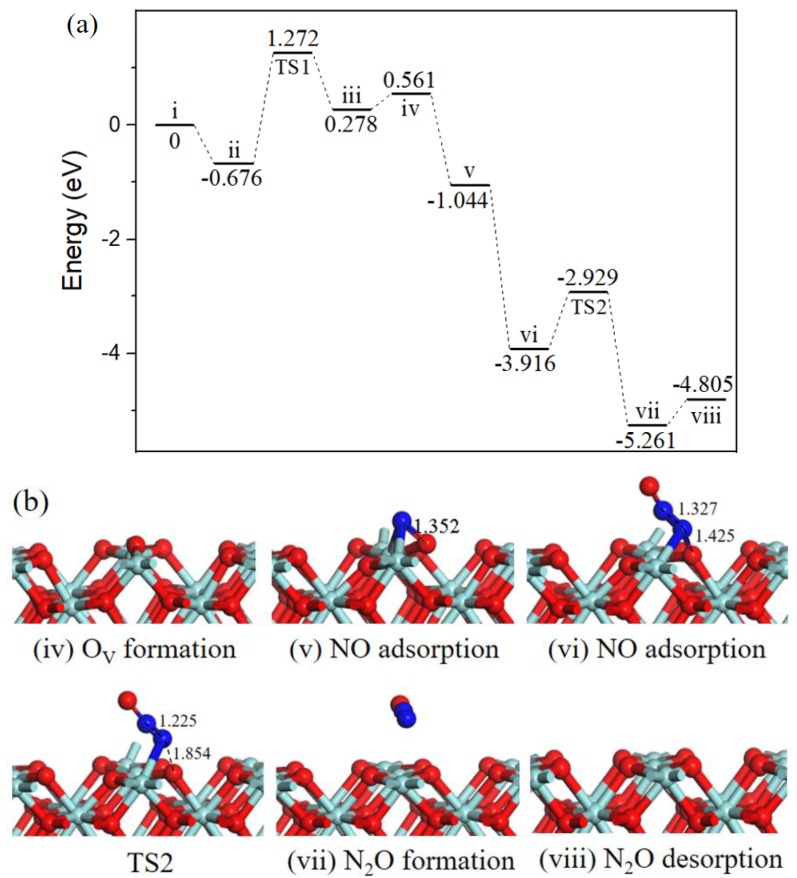
Energy profile (**a**) and structure models (**b**) of reactants, transition states and products for N_2_O formation on ZrO_2_ (110) surface. The energy profile is placed above, and structure models are placed below.

**Figure 5 ijms-20-06129-f005:**
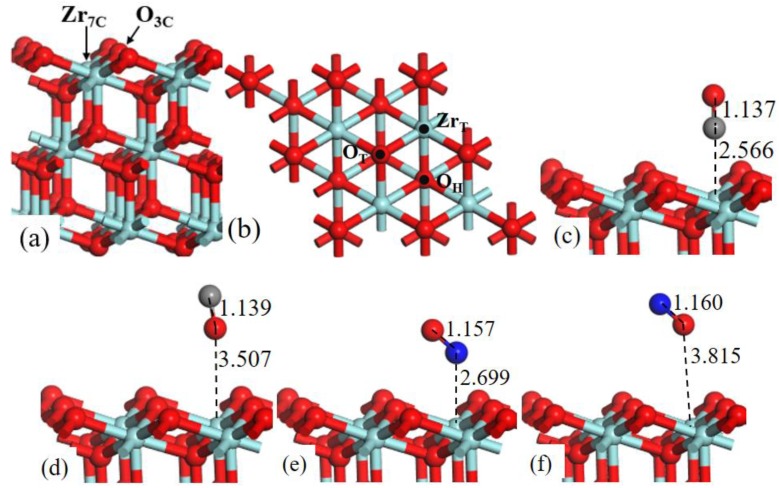
Structure models of (**a**) the side view and (**b**) corresponding top view of ZrO_2_ (111)-2 × 2 surface. (b) Three possible adsorption sites for CO and NO gas molecules are labeled. Optimized adsorption structures of (**c**) CO C-end, (**d**) CO O-end, (**e**) NO N-end, and (**f**) NO O-end on Zr_T_ site.

**Figure 6 ijms-20-06129-f006:**
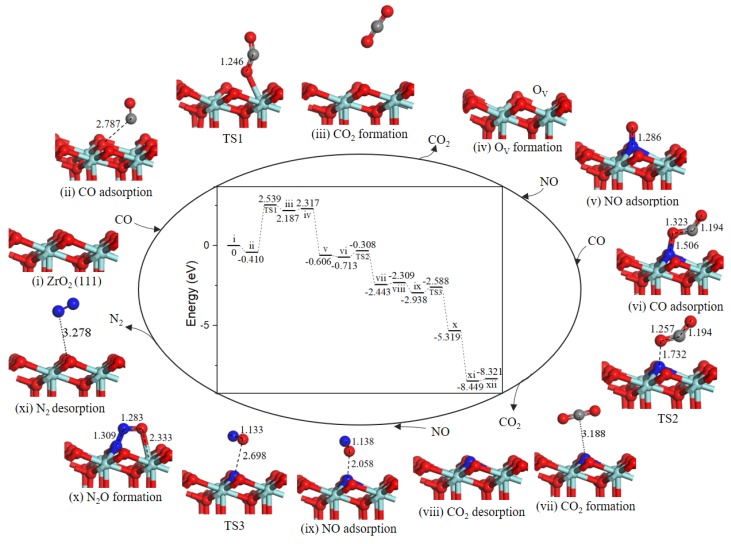
Energy profile and structure models of reactants, transition states and products for path 1’ (NO N-end embed in oxygen vacancy site on ZrO_2_ (111) surface). The energy profile is placed in the center, and structure models are placed around.

**Figure 7 ijms-20-06129-f007:**
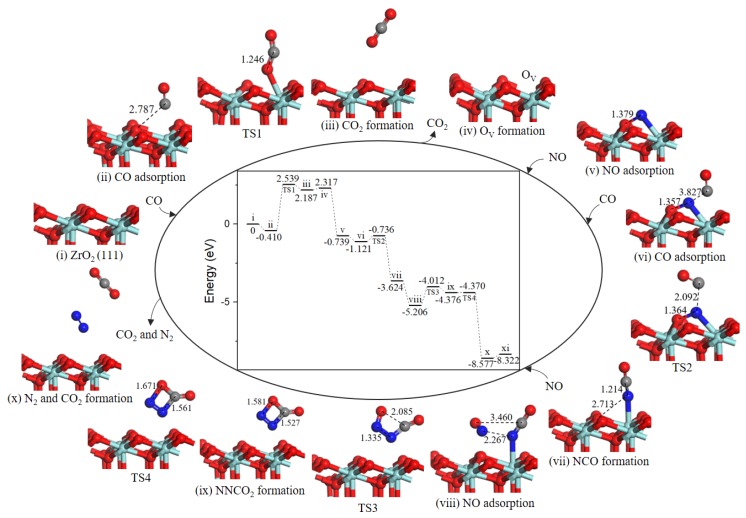
Energy profile and structure models of reactants, transition states and products for path 2’ (NO O-end embed in oxygen vacancy site on ZrO_2_ (111) surface). The energy profile is placed in the center, and structure models are placed around.

**Figure 8 ijms-20-06129-f008:**
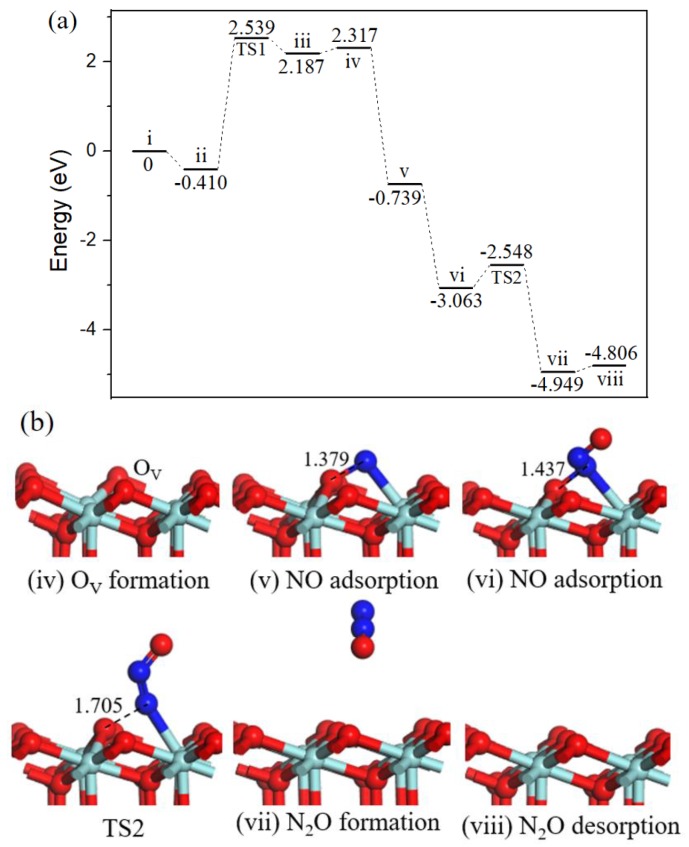
Energy profile (**a**) and structure models (**b**) of reactants, transition states, and products for N_2_O formation on ZrO_2_ (111) surface. The energy profile is placed above, and structure models are placed below.

**Figure 9 ijms-20-06129-f009:**
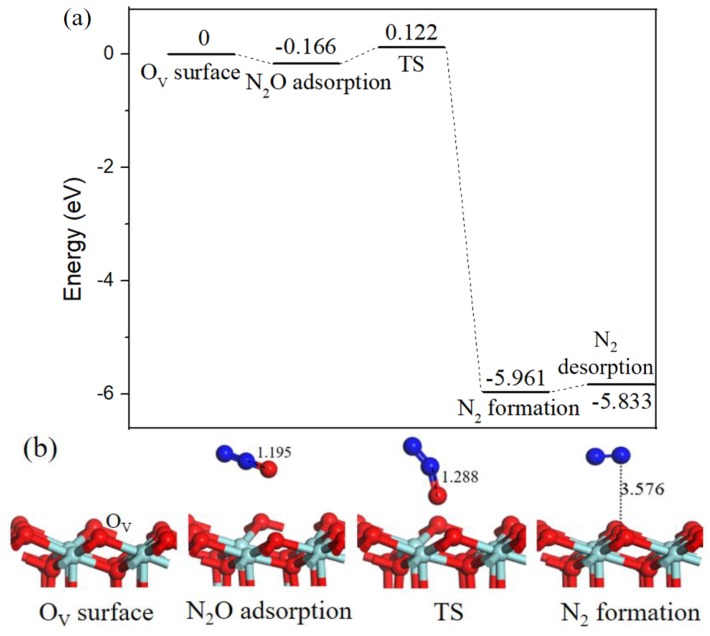
Energy profile (**a**) and structure models (**b**) of reactants, transition states and products for N_2_O adsorb on oxygen vacancy ZrO_2_ (111) surface to generate N_2_ (g). The energy profile is placed above, and structure models are placed below.

**Table 1 ijms-20-06129-t001:** Adsorption energies (*E*_ads_) of CO and NO gas molecules on the different adsorption sites of ZrO_2_ (110) surface.

Gas Molecule		Adsorption Site	*E*_ads_ (eV)	Figure
CO	C-end	Zr_T_	−0.676	1(c)
		O_T_	−0.675	-
		O_b_	−0.227	-
	O-end	Zr_T_	−0.359	1(d)
		O_T_	−0.257	-
		O_b_	−0.231	-
NO	N-end	Zr_T_	−0.788	1(e)
		O_T_	−0.776	-
		O_b_	−0.253	-
	O-end	Zr_T_	−0.575	1(f)
		O_T_	−0.290	-
		O_b_	−0.047	-

**Table 2 ijms-20-06129-t002:** Adsorption energies (*E*_ads_) of CO and NO gas molecules on the different adsorption sites of ZrO_2_ (111) surface.

Gas Molecule		Adsorption site	*E*_ads_ (eV)	Figure
CO	C-end	Zr_T_	−0.410	5(c)
		O_T_	−0.129	-
		O_H_	−0.160	-
	O-end	Zr_T_	−0.162	5(d)
		O_T_	−0.133	-
		O_H_	−0.154	-
NO	N-end	Zr_T_	−0.041	5(e)
		O_T_	0.078	-
		O_H_	0.046	-
	O-end	Zr_T_	0.054	5(f)
		O_T_	0.076	-
		O_H_	0.055	-

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
