# Peer review of "Catalytic Reaction Mechanism of NO–CO on the ZrO2 (110) and (111) Surfaces"

_ijms, 2019, doi:10.3390/ijms20246129_

Round 1

Reviewer 1 Report

In the manuscript "Catalytic reaction mechanism of NO-CO on the ZrO2 (110) and (111)", authors present the results of a series of DFT calculations on the oxidation of CO to CO2 through Mars-van Krevelen mechanism catalyzed by ZrO2, where NO is reduced instead of O2.

The manuscript is physically sound and provides valuable insights in the reaction mechanism. The setup of DFT calculations is adequate and conclusions are supported by data. The text is not sufficiently well-written and needs extended editing.

I have only one remark regarding adsorption energies:

In tables 1 and 2, adsorption energies of CO and NO on several sites of ZrO2 (110) and (111) surfaces are presented.

In all the reported cases, several adsorption configurations collapse on another during the geometry optimization. For instance, in the adsorption of CO on ZrO2(110) through the C-end, Zr-O, OH, OT, Ob1, and Ob3 adsorption sites are unstable and in all these cases the CO molecule ends up on top of a Zr atom (ZrT) at the end of the geometry optimization. Reporting an adsorption energy for those configurations is both meaningless and misleading. All of them should be regarded as unstable. Besides, differences in the order of 10^-3 eV are meaningless in a DFT calculations, which has an expected accuracy at least one order of magnitude larger.

I strongly suggest to revise tables 1 and 2 and correctly address.

I also suggest to improve the readability by separating the data referring to the adsorption through the O-end from the ones referring to C- and N-ends with a horizontal line.

Reviewer 2 Report

The paper investigated reported the catalytic reaction mechanisms in presence of NO and CO on ZrO2 surfaces. Different surfaces (110) and (111) are taken into account. The authors reported several DFT mechanisms and explored in details the complex steps of the mechanisms. The major doubt is the complete absence of experimental data on this system, also from literature. As they suggested, the calculated energy barriers, especially for the TS1 in CO2 formation in ZnO2(110) and (111), are very high. This prevent the possibility to use these mechanisms at room temperature. At page 5, they cite Carlotto et al. JPCC 2018, where the importance of the dopant is highlight and in particular is reported the incapacity of the no-doped perovskite to act as catalyst. This behavior is probably due to the high energy barrier for oxygen vacancy formation (below 1.5 eV). Analogously, in ZrO2 systems, the energy barriers for the oxygen vacancy formation are higher than 1 eV, hence I supposed, also supported by the literature [Carlotto et al. JPCC 2018, Carlotto PCCP 2016, 18, 33282], that these systems (not doped) cannot act as catalyst (except, maybe, at very high temperature). The study is very interesting, but in absence of an experimental comparison (also from literature), it is difficult to believe that the material can be a catalyst. Major revisions are necessary to convince this reviewer that ZrO2 is a catalyst for CO oxidation and NO reduction.

Reviewer 3 Report

Cao et al. present an extensive DFT study of NO reduction/CO oxidation over two typical ZrO2 surfaces.

Overall, the paper is constructed in a repetitious way, i.e., the pathways and numbers are extensively described without much discussion of the meaning of the result. Also, the comparison between the two surfaces is far from obvious to get when reading the paper: No Figure or paragraph is dedicated to this aspect.

Another missing point is the discussion of the surface model: Are ZrO2(110) and ZrO2(111) low-energy surfaces? - Is one much lower in surface free energy than the other? Are there other low-energy surfaces of zirconia? What about the surface state under realistic conditions? - Should there be more/less oxygen compared to the stoichiometric surface?

The method applied is likely to be rather imprecise. To validate their setup, the authors should compare overall reaction energetics as computed by DFT to NIST standard values and, if available, temperature-programmed desorption of NO or CO over zirconia could also give some valuable information.

To clarify the relevance of their values, the authors should also provide energetic spans as derived from their catalytic cycles. These values can then be converted into approximate rates, which can be more directly compared to experiments. In order to do so, the authors should also include free energy corrections, at least for the gas-phase species. In view of the high barriers and the low activity of ZrO2 alone (in the introduction it is not entirely clear if ZrO2 alone shows at least a small, high-temperature activity) for the studied reactions, the relevance of the study should be clarified.

The clarity of the Figures needs to be improved: There are many numbers that overlap with the pictures.

The authors should provide the coordinates of all optimized structures in the supplementary information.

Energy values should not be quoted to meV precision. Even tenths of eV are enough for the content of this study.

Round 2

Reviewer 2 Report

The authors partially answer to my questions. They add experimental comparison from literature to demonstrate that ZrO2 is a catalyst for CO oxidation and NO reduction and this is a good improvement, but the major problem still remain. They presented different paths with different TSs. In both paths (1 and 2), the initial steps have barriers around 2 eV. At mild conditions, these paths are not thermodynamically favor, hence the proposed mechanisms improbably can be observed. The reported literature supported this idea. At the present, authors persuade me that the ZrO2 system is a catalyst for this reaction, but the proposed mechanisms have too high barriers. To have my definitive approval for publication, authors should include some studies to demonstrate that, for this system, these energy barriers are acceptable.

Reviewer 3 Report

The revision of the manuscript has brought some improvements, but major points have not been adressed:

The energetic span is not computed, even though it just requires some simple evaluations that can be easily done via Excel The Figures still feature overlaping text (bond lengths) and graphical representations The experimental motivation is lacking: the experiments are performed on zirconia decorated with copper.  The role of copper is certainly crucial but completely neglected in the present manuscript.

While the first two points are minor, the last one is probably beyond the scope of the manuscript. Hence, the manuscript can be rejected (lack of relevant insight) or accepted (technically sound data) at the discretion of the editor.

Round 3

Reviewer 2 Report

The authors answered all the questions raised by my reviewer in a proper way. The manuscript text and the references were revised in accordance with the comments of the reviewer. In this way, the overall presentation of the findings improved. The manuscript can now be published as is.